# Evaluating Fairness and Mitigating Bias in Machine Learning: A Novel Technique using Tensor Data and Bayesian Regression

## Abstract

Fairness is a critical component of Trustworthy AI. In this paper, we focus on Machine Learning (ML) and the performance of model predictions when dealing with skin color. Unlike other sensitive attributes, the nature of skin color differs significantly. In computer vision, skin color is represented as tensor data rather than categorical values or single numerical points. However, much of the research on fairness across sensitive groups has focused on categorical features such as gender and race. This paper introduces a new technique for evaluating fairness in ML for image classification tasks, specifically without the use of annotation. To address the limitations of prior work, we handle tensor data, like skin color, without classifying it rigidly. Instead, we convert it into probability distributions and apply statistical distance measures. This novel approach allows us to capture fine-grained nuances in fairness both within and across what would traditionally be considered distinct groups. Additionally, we propose an innovative training method to mitigate the latent biases present in conventional skin tone categorization. This method leverages color distance estimates calculated through Bayesian regression with polynomial functions, ensuring a more nuanced and equitable treatment of skin color in ML models.

## 1 Introduction

Machine Learning (ML) is gaining widespread use across various domains, potentially influencing society profoundly. Accordingly, attention has turned towards the risks associated with ML. A significant risk to consider is unfairness towards ethnic and other social groups. A particular case of this risk is unfairness in the predictive performance of deep-learning image classification models, e.g. for cancer detection, depending on skin color Lin et al. (2024); Muthukumar (2019); Buolamwini & Gebru (2018); Bevan & Atapour-Abarghouei (2022); Pakzad et al. (2022); Sarridis et al. (2023). Prior studies have contributed to the consensus that ML classifiers perform poorly on darker skin tones and better on lighter skin tones. Skin color is a well-recognized protected characteristic that should not be discriminated against under emerging guidelines legislation.gov.uk (2013) on AI safety. Skin color is one the harder sensitive attributes to address in research of AI fairness. There are two key difficulties.

The first is the difficulty in achieving consistency in objective judgments of skin color. Experts have not achieved complete agreement on skin color grouping in previous studies Groh et al. (2022); Krishnapriya et al. (2021); Heldreth et al. (2024). There are numerous skin color scales Thong et al. (2023), such as the **?** validity and Monk skin scales Schumann et al. (2024), but there is still no established method for identifying a single definitive skin color categorization. Moreover, the grouping of skin color is not determined exclusively by its color. It is frequently substituted for ethnic groups, such as Black, White and Asian. While race is classified according to physical characteristics, ethnicity is determined by an individual's background Bulatao & Anderson (2004). Considering the increase in diversity in modern society, the racial characteristics of traditional ethnic groups can not necessarily be represented. Research indicated that individuals selected their ethnicity, taking into account the context. Therefore, whether an individual's skin color is light or dark is a subjective judgment, and there is the possibility that biases caused by category selection may be hidden.

The second problem is the data value attribute of skin color. Many protected attributes are categorical values, such as sex. For example, $\mathbb{A} = \{\text{male}, \text{female}\}$ is a sensitive attribute that can only take one of the categorical values in the defined set, and this can be done using judgment based on well-defined criteria. Another type of protected attribute, such as age, takes single numerical data, $\mathbb{A} = \{1, .., n\}$. Such attributes are given a single data value in tabular data or annotations. In recent years, methods for assessing fairness and mitigating biases corresponding to sensitive attributes with continuous numerical data have emerged Mary et al. (2019); Grari et al. (2019); Giuliani et al. (2023); Brotto et al. (2024); Lee et al. (2022); Grari et al. (2023); Oneto et al. (2020). However, these studies focus on simple tabular numerical data, and such data is intrinsically different from image data Tian et al. (2022). Skin color does not fit easily into studied these categories. Skin color is tensor data in computer vision and is represented as the set of each pixel in the skin area, represented with values for each of the three primary colors. Nevertheless, most previous research on ML biases on skin colour has assumed traditional group classification. The differences between the same group are fundamentally ignored Chouldechova & Roth (2018). Categorization involves and amplifies the risk of uncertainty by statistically averagingRuggieri et al. (2023).

Furthermore, large parts of research demand skin color type annotation on image data. This requires a great deal of effort and annotation accuracy is critical Kalb et al. (2023). A classification method that included skin color differences without annotations was proposed, but this was based on transfer learning, and annotations were still used for the source model Hwang et al. (2020). To our knowledge, no research has achieved a fair model without annotations using only detected skin color nuances. The primary factor contributing to bias is the imbalance in the distribution of skin tones in available datasets. Hence, several studies also focus on creating balanced datasets Gustafson et al. (2023); Karkkainen & Joo (2021).

Motivated by the above, we propose a method for measuring skin color to assess individual fairness for skin color within and across subgroups. Unlike previous methods, this method converts skin pixels tensor data to a probability distribution. It then uses a statistical distance to measure the differences in the probability distribution of each individual's skin color while maintaining the gradation and color nuances of the skin. The method enables the detection of skin color bias that has previously been masked within groups, and the identification of biases that have not been detected due to the lack of annotations. Furthermore, we propose a method of weighting the loss function by the distance to mitigate the bias detected by our method. This method reduces the correlation between skin color distribution and performance.

## 2 RELATED WORK

We focus on image classification focusing on skin color that affects fairness towards racial or ethnic groups. Generative image, facial recognition, and object segmentation tasks are out of the scope. Earlier studies have shown that bias arises from the limited number of images available for darker skin tones. **Generative Adversarial Networks (GAN)** have therefore been used to balance the dataset by oversampling images with minority skin tones Rezk et al. (2022). Another method was to generate counterfactual data of minority skin tones Li & Abd-Almageed (2022); Dash et al. (2022). These methods generally require the same effort as creating balanced datasets. Another approach to the detection of skin cancer with ML is **Removal or Compliment**. The method removed sensitive attributes. Chiu et al. (2024) proposed a technique for skin lesion classification that classifies the type of disease based only on features related to the target attributes and does not distinguish features associated with the sensitive attribute, which is skin color. A method was proposed for clinical skin image data that takes into account differences in skin tone and aligns with the text data and with the Masked Graph Optimal Transport subsequently denoised Gaddey et al. (2024). Lee et al. (2021) et al. proposed selective classification. These methods succeed in specific datasets and conditions, but they cannot apply to general skin datasets. Other relevant research focused on the application of **Explainability techniques**. Wu et al. (2022) performed saliency calculations and reduced disparities between groups by averaging out the importance of the parameters for each skin-color group. Cross-Layer Mutual Attention Learning mitigated bias by complementing the features of deep layers with the color features found in shallow layers Manzoor & Rattani (2024). These methods compared the differences between groups of features that the model focused on during the prediction process and ignored the disparities in skin color between individuals. **Adversarial learning** separates the sensitive attributes during learning to prevent the model from learning sensitive attribute features Li

et al. (2021); Du et al. (2022); Park et al. (2022); Wang et al. (2022); Bevan & Atapour-Abarghouei (2022). In an application for Deep Fake detection, demographic information, including protected attributes and fake features, was separately trained and merged to optimize the loss Lin et al. (2024). All of these methods tend to result in relatively complex model structures. **Fairness-constrained and Reweighing learning** was applied with a weighted loss function using weighted cross-entropy to mitigate bias Hänel et al. (2022). Our bias mitigation technique is also categorized into this concept, but the reweighing methodology is fundamentally different. Ju et al. (2024) et al. have proposed a demographic-agnostic Fair Deepfake Detection that minimizes the error for the worst performance by group creating a new loss function to guarantee fairness even when annotations for sensitive attribute groups are missing . Lin et al. (2022) proposed a method for balancing the importance of weights within a model for subgroups in the pruning process. Thong & Snoek (2021) et al. used a latent vector space to remove the bias from the image. Another approach developed Q-learning in reinforcement learning to minimize bias by setting rewards according to the skewness in class distance between races Wang & Deng (2020). A bias removal by converting an image into a sketch kept the features for the model decisionYao et al. (2022). Zhang et al. (2022) et al. proposed a fairness trigger to add biased information to images. By clarifying the edge of the skin lesions, the difference in accuracy between light-skinned and dark-skinned samples was eliminated Yuan et al. (2022). In the implementation of fair image classification for skin tones, various algorithms, such as those mentioned above, have been proposed. Nevertheless, there is a commonality among all these studies that they categorize or assume grouping skin tone. Therefore, potential biases may still remain in those mitigation systems. The finer characteristics of skin should be taken into account. To address these challenges with the existing fairness evaluation and unfairness mitigation approach, we propose a new statistical-based approach and weighted loss function learning with the following main contributions:

1. In the context of skin color image classification tasks, we propose an innovative algorithm to evaluate more nuanced individual fairness within group fairness without annotation and by using statistical distance and Bayesian regression.

2. We demonstrate the ability to uncover latent bias within categorization using our method.

3. We propose a new training method to mitigate latent bias across the spectrum of skin color variation, creating a new weighted loss function by weight cross-entropy.

4. We evaluate the effectiveness of the training method in mitigating latent bias.

5. We make all code for the above publicly available for further work and experiments by third parties. Anonymized repository (https://anonymous.4open.science/r/FairSkinColor-D910/)

## 3 METHODOLOGY

Figure 1 illustrates the learning method for the proposed bias mitigation. This learning method is divided into two processes. The first process is the prior learning process, which includes general training and skin color measures. An image from each dataset is selected as the baseline skin color distribution for validation, as the validation performance is used as prior data for Bayesian regression. Then, the distance between the color differences of all other validation data is measured from the baseline color. The process of measuring skin color is explained in detail in the following subsections. A performance estimator model is created by fitting the results and validation predictions using Bayesian regression. This estimator is assembled during the second process, known as posterior training. The posterior training applies a weighted loss function that penalizes the inverse of predictive distance performance.

### 3.1 SKIN COLOUR IDENTIFIER

Our method aims to preserve the nuances of pigmentation inherent in skin tones. In computer vision, skin color in color images comprises three pigments across three channels per pixel. In our study, to align with human perception for real-world applicability and enable direct comparison with categorical skin types used in previous research, we adopt the Individual Typology Angle (ITA). ITA is frequently used for skin color fairness studies as the foundation of representative skin colors Kinyanjui et al. (2019); Corbin & Marques (2023); Kalb et al. (2023); Mohamed et al. (2023). However, these studies treat ITA values as a single numerical value, representing a single continuous numeric

Figure 1: Bias Mitigation Learning Process: The performance estimator for the posterior training is a Bayesian regression created in the prior training phase. In the posterior training, the skin color of the training data is measured. The base skin color is the same as the validation data. One of two types of loss functions is applied depending on the epoch.

sensitive attribute group. In extant research the nuances of skin tone pixels are not considered; instead, they are averaged out. Furthermore, even the ITA values themselves are not retained to measure fairness; they are replaced with categorical values. This results in disregarding the inherent properties of skin color. ITA is calculated in the CIELab color space according to the following equation for ITA in algorithm 1. $L$ and $b$ are defined as Lightness and b-hue.

---

**Algorithm 1** Skin Colour Identifier: Creating Skin Nuance Color Distribution

---

**Input**: Image $x \in \mathbb{R}^{w \cdot h \cdot 3}$
**Output**: Nuance Skin Colour Distribution $\boldsymbol{v}$
  1: $\mathbf{S} = SkinDetector(x)$ {Selected based on the dataset type.}
  2: $\boldsymbol{L}, \boldsymbol{A}, \boldsymbol{B} = CIELab(\mathbf{S})$ {Convert to CIELab color space.}
  3: $i = 0, j = 0$
  4: **for** $i < \boldsymbol{L}^h$ **do**
  5:    **for** $j < \boldsymbol{L}^w$ **do**
  6:      $l = L_{i,j}$
  7:      $b = B_{i,j}$
  8:      $j = j + 1$
  9:      **if** $l \neq 0 \cap b \neq 0$ **then**
10:        $ITA = \frac{\arctan\left(\frac{L-50}{b}\right) \times 180}{\pi}$
11:        $\boldsymbol{v} = \boldsymbol{v} + ITA$
12:      **end if**
13:    **end for**
14:    $i = i + 1$
15: **end for**
16: **return** $\boldsymbol{v}$

---

### 3.1.1 MEASURING SKIN COLOUR DISTANCE

Assuming the distributions are IID, the Wasserstein Distance (WD) is recognized as one of the best approaches for capturing changes in the geometry of the distribution, effectively highlighting shifts that reflect underlying data transformations Cai & Lim (2022). The WD effectively quantifies the minimum effort required to reconfigure one distribution into another, which measures the variability of skin color shades across images in this context. Specifically, the baseline image, denoted by $\boldsymbol{x}_0$, is selected randomly from the validation dataset, serving as the reference distribution. Subsequent distributions, represented by $\boldsymbol{x}_i$ where $i$ indexes these distributions, are compared against $\boldsymbol{x}_0$ using the Wasserstein metric. This metric assesses the extent to which the skin color distributions shift towards lighter or darker tones, assigning a quantitative measure that reflects the minimal cost of transport from the baseline to each observed distribution. The sign function, $\mathcal{S}(\boldsymbol{x}_0, \boldsymbol{x}_i)$ is defined as

follows:

$$Sign = \mathcal{S}\left(\boldsymbol{x}_0, \boldsymbol{x}_i\right) = \left\{ \begin{array}{ll} -1 & : \; median\left(\boldsymbol{x}_0\right) \geq median\left(\boldsymbol{x}_i\right) \\ 1 & : \; median\left(\boldsymbol{x}_0\right) < median\left(\boldsymbol{x}_i\right) \end{array} \right. \tag{1}$$

Then, the values measured by WD are multiplied by the sign to quantify the difference between skin tones and their saturation direction.

$$Distance = \mathcal{D}\left(\boldsymbol{x}_0, \boldsymbol{x}_i\right) = \int \left|\mathcal{F}\left(\boldsymbol{x}_0\right) - \mathcal{F}\left(\boldsymbol{x}_i\right)\right| dx \cdot sign \tag{2}$$

## 3.2 Performance Estimation Bayesian Regression Model

Since our techniques are designed for binary classification, where individual predictions are either 0 or 1, performance cannot be effectively measured at the individual level. To address this, batches are created by small groups of similar distances after sorting in ascending order of the $\mathcal{D}\left(\boldsymbol{x}_0, \boldsymbol{x}_i\right)$. The batch size was set to 1% of the validation dataset, allowing for a more accurate assessment of performance in the experiment. The technique uses Bayesian Regression to predict performance using generic models from skin tones. Let $D = \{d_0, ..., d_{n-1}\}^T$ denote the vector representing the distance from baseline skin color as measured by the distance function above. The performance associated with distance is $M = \{m_0, ..., m_{n-1}\}$ where $n$ is the number of instances. The visualisation of the observed performance suggests that the regression model assumed polynomial features. The degree of the polynomial regression depends on the model and dataset and is determined from the prior distribution. The degree denotes $g$.

$$D = \begin{bmatrix} d_0 & d_0^2 & \cdots & d_0^{g-1} \\ d_1 & d_1^2 & \cdots & d_1^{g-1} \\ \vdots & \vdots & \ddots & \vdots \\ d_{n-1} & d_{n-1}^2 & \cdots & d_{n-1}^{g-1} \end{bmatrix} \tag{3}$$

The prior distribution $p\left(M|D, w, \alpha\right)$, follows the Gaussian Distribution, $\mathcal{N}\left(M|D_w^g, \alpha^{-1}\right)$. $\boldsymbol{w}$, and $\alpha^{-1}$ are, respectively, the coefficients and the precision. The coefficients $\boldsymbol{w}$ are provided by Spherical Gaussian: $p\left(w|\lambda\right) = \mathcal{N}\left(\mu, \lambda^{-1}\mathrm{I}_p\right)$, where $\mu$ is mean and set 0. Given the distance $D_{test} = \{d_0, ..., d_{n-1}\}^T$ of the new test data $X_{test}$, the likelihood of the prediction performance $\hat{M}_{test} = \{m_0, ..., m_{n-1}\}$ is calculated $\mathcal{P}\left(m|d\right)$ using the following equation.

$$\hat{M}_{test} = \mathbb{E}\left[m\right] = \int m p\left(m|p\right) dm \tag{4}$$

## 3.3 Latent Bias Mitigation

The binary cross entropy loss function is used to guide bias mitigation. The individual loss $l$ is formulated as follows.The penalty value assigned to the binary cross entropy loss is calculated by weighting and averaging the prediction performance inversion using the softmax function, $\sigma\left(1 - \varepsilon\right)_i = \frac{e^{(1-\varepsilon)_i}}{\sum_{j=1}^{K} e^{(1-\varepsilon)_j}}, . \; .$, where $\left(1 - \varepsilon\right)$ denotes the penalty, and $\varepsilon$ is performance prediction calculated based on skin color probability distribution distance by the Bayesian Regression Estimator equation. Since the convolutional neural network-based model gradually focuses on more detailed features in the learning process, it is unnecessary to penalise the nuanced features of the skin in the early stages of learning. Therefore, only the binary cross-entropy value is applied until the middle of the process, and weighting is performed after that. $\alpha$ is a penalty weight. The entire Loss function is algorithm 2.

$$l_n = -w\left\{y_n \cdot \log x_n + \left(1 - y_n\right) \cdot \log x_n\right\} \cdot \sigma \cdot \alpha \tag{5}$$

---

**Algorithm 2** Distance Loss Function: Calculate loss function with distance penalty

---

**Input**: Prediction $\hat{y}$, Target Label $y$, Distance $d$, Penalty Epoch $pe$, Epoch $e$, Batch size $N$
**Output**: Loss $l$

1: Initialize BCE
2: **for** $n < N$ **do**
3:     $bce = \text{BinaryCrossEntropy}(\text{Sigmoid}(\hat{y}), y)$
4:     BCE = BCE + bce
5: **end for**
6: **if** $e \le pe$ **then**
7:     $l = \frac{1}{N} \sum_{n=1}^{N} BCE_n$
8: **else**
9:     $\varepsilon = \text{BayesianPerformanceEstimator}(d)$
10:    Penalty $p = \text{Softmax}(1 - \varepsilon)$
11:    $l = \sum_{n=1}^{N} BCE_n \cdot p_n \cdot \alpha$
12: **end if**
13: **return** $l$

---

### 3.4 SELECTING PERFORMANCE EVALUATION METRICS

Our technique detects and mitigates the latent bias caused by individual skin tone categorisation. It focuses on ensuring individual fairness using the skin tone spectrums. Therefore, we do not evaluate our technique on group-level fairness metrics such as Demographic Parity Zafar et al. (2017), Equalised Odds and Equal Opportunity Hardt et al. (2016), which are commonly used in studies that categorise skin tones. Since our model is mitigating bias on an individual level, it's important to reduce both false positives (cases where an individual's skin tone is misclassified) and false negatives (where bias is not detected). The F1 score provides a balanced view of both types of errors, especially useful when classes (or skin tones) are imbalanced, which can easily happen in skin tone data. Consequently, the F1 score and Accuracy are selected as the evaluation metrics to focus on. The proposed mathematical formulations of concepts for Equal Opportunity, Demographic Parity, and Equalized Odds for continuous attributes are given in Appendix B.

## 4 EXPERIMENTAL SETUP: DATASETS AND MODELS

### 4.1 DATASETS AND SKIN DETECTION

The following three types of datasets were adopted. Each dataset was divided into a training set (60%), a validation set (20%), and a test set (20%). The training datasets were balanced in targeting labels. Then, the number of images between the skin color types was also equalized to simulate a state where statistical fairness was ensured between the subgroups in the training dataset. The detailed breakdown of the datasets is shown in the table in the appendix. Different approaches were employed to detect skin depending on the dataset because the background conditions for skin pixels differ. The details are shown as follows.

1. Human Against Machine with 10000 training images (HAM): This is a training dataset for skin lesion classification collected from dermatoscopic images acquired and stored by different modalities from different populations Tschandl et al. (2018; 2020). **Skin Detection:** Skin color identification was conducted using publicly available lesion segment images Tschandl et al. (2018). **Skin Color Category:** The skin color was classified into Fitzpatrick skin color categories based on the mean of the ITAs using conventional methods. Only the largest number of skin-tone type 1 was used in the experiment. It is possible to ascertain whether there are performance differences by skin nuance within a single skin color type.

2. CelebFaces Attributes Dataset (CelebA): CelebA is a sizeable facial attribute dataset containing over 200K celebrity images with 40 diverse attribute annotations Liu et al. (2015). **Skin Color Category:** In this dataset, skin tones are binary classified as pale or not. **Skin Detection:** The facial recognition landmark method recognized the face, eyes, and mouth

Table 1: Experiment dataset and skin detection methods

| Dataset | UTKFace | CelebA | HAM |
|---|---|---|---|
| Classification Tasks | Gender | Face attribute | Skin lesion |
| Category | Ethnic Group | Pail or nor | Fitzpatrick Skin Type1 |
| Skin Detection | Landmark | Landmark | Segmentation |
| Target | Male or Female | Positive or Negative | Melanocytic Nevi or Melanoma |
| Train Total (n) | 7133 | 6426 | 1300 |
| Train Class 0 (n) | 3546 (928, 844, 871, 903) | 3161 (1623, 1538) | 650 |
| Train Class 1 (n) | 3587 (884, 889, 886, 928) | 3265 (1649, 1616) | 650 |
| Validation (n) | 2348 | 2134 | 434 |
| Test (n) | 2348 | 2129 | 434 |

King (2009). The non-face areas, including the eyes and above the top of the eyes and mouth, were then masked. Images for which face recognition was not possible, such as side view of faces, were excluded.

3. UTKFace: This is a sizeable facial dataset with a wide age range, consisting of more than 20,000 face images annotated with age, gender, and ethnicity Zhang et al. (2017). **Skin Color Category:** This dataset was chosen because skin tones are often categorized by ethnic group. Race is sometimes used to contextualize or identify with skin color Barrett et al. (2023). **Skin Detection:** Skin color detection was conducted using the same method as the CelebA dataset.

Details and summaries of the dataset after pre-processing had been carried out are in the following Table 1. The values in the brackets for Train Classes are the number of data for each categorical skin type. The skin color groups were balanced with a maximum difference of 5%. Group fairness was achieved.

## 4.2 MODELS

Three pre-trained models using the ImageNet dataset, Very Deep Convolutional Networks (VGG16) Simonyan & Zisserman (2014), EfficientNet7B (EffNet) Tan & Le (2019), and ResNet50 He et al. (2016), were selected for this experiment. All are based on convolutional networks and are commonly used in image classification tasks. Since the data set was undersampled to create balanced subcategories, reducing the number of available images for training, pre-trained models were incorporated. This approach ensures that good performance can still be achieved, even with a limited amount of training data. Each model was additionally trained for each dataset. The general performance and training conditions are shown in Table 4 below. As can be seen from Table 4, the general prediction results demonstrated that the models did not differ significantly in performance based on skin color tone.

## 5 RESULTS

In this section, we describe the results of the experiment. Figure 2 illustrates the ten samples extracted from the UTKFACE dataset. All of these samples are face images annotated as 'white' skin color. Image (A) shows the original image with added landmarks in red. Image (B) shows the skin area in the face extracted by the landmarks, with the non-skin color areas masked in black. From these images, it is evident that the skin color gradation differs from each face when viewed by human eyes. Figure (C) plots the probability distribution of the pixels of only the skin color area of (B). In this figure, the visual nuance differences of the image in (B) can be expressed numerically.

Figure 3 is a performance prediction Bayesian regression model fitted using the validation data as a prior and general model. The blue plots employed the F1 score as the metric, and the green plots show Accuracy. The red horizontal line shows the mean score for the validation dataset. The grey scatter plot provides the prior observed data. The value on the X-axis is 0 for the base sample. The lighter skin colors are the larger values, and the darker are the greater negative values. In the case of the weaker correlation between skin color and performance, the Bayesian regression performance estimator, such as CelebA, is flatter. Conversely, UTKFace and HAM tend to have

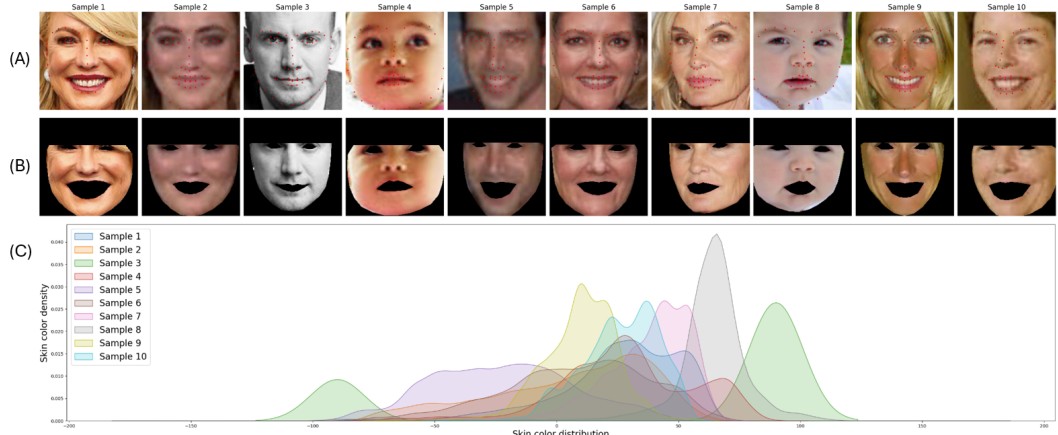

Figure 2: Examples for skin gradation distribution: (A) These are the original image and the landmark of the 10 UTKFACE samples. (B) These are images in which only the skin pixels have been extracted by masking out all pixels except for the skin pixels. (C) is a probability distribution of the ITA values calculated for each skin pixel.

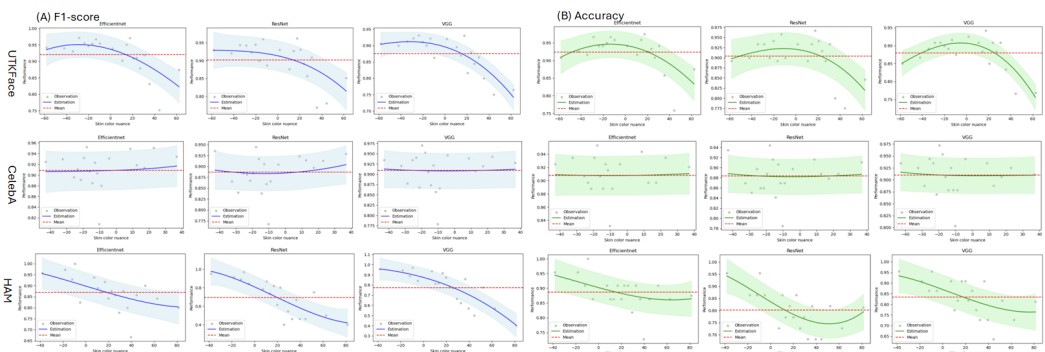

Figure 3: Bayesian Performance Estimators: This shows the performance prediction of the Bayesian regression model using the validation dataset as prior for each model and dataset. The blue graph (A) is a prediction model based on the F1 score, and the green graph (B) is based on accuracy.

apparent differences depending on skin color. This shows that the element of skin color has an enormous impact on the model's predictions. The observed individuals of predictions that are below the average of the prior are due to their skin spectrum. Next, the results of the posterior training process of incorporating the model that predicts the change in F1 score according to the skin tone displayed in Figure 3 into the loss function are shown in Table 2. Table 3 shows the correlation between the performance of each evaluation metric and distance when the batch size is 1%. In the prior training, a negative correlation with the F1 score was shown in the UTKFace and HAM datasets. The performance deteriorated as the color gradient became lighter. In the HAM dataset, a correlation was also observed in the Eff and ResNet accuracy. In the CelebA dataset, no correlation was provided in any of the models. This is because the skin color in this dataset was centred around the median compared to the others.

The results of the posterior-training bias mitigation are shown on the right side of Table 3. In most cases of the combination of the models and datasets, the correlations between distance F1 score and accuracy were mitigated. The CelebA, which originally showed no correlation, also had relatively decreased coefficients. In the case of the UTKFace dataset and models of Efficientnet and Resnet, the weak correlation was no longer observed. Regarding the HAM and Efficientnet combination, the moderate correlation was mitigated toward a weak correlation.

Table 2: Posterior training performance results

| Dataset | UTKFace | | | CelebA | | | HAM | | |
|---|---|---|---|---|---|---|---|---|---|
| Model | Eff | ResNet | VGG | Eff | ResNet | VGG | Eff | ResNet | VGG |
| lr | 1e-5 | 1e-5 | 1e-6 | 1e-6 | 1e-6 | 1e-6 | 1e-5 | 1e-5 | 1e-6 |
| Epochs | 23 | 23 | 19 | 29 | 28 | 24 | 23 | 19 | 12 |
| Penalty Start | 16 | 17 | 1 | 17 | 17 | 17 | 12 | 18 | 15 |
| Penalty Weight | 0.95 | 1 | 0.95 | 1 | 1 | 1 | 0.95 | 0.95 | 1 |
| Val F1 | 0.89 | 0.91 | 0.88 | 0.91 | 0.88 | 0.92 | 0.90 | 0.87 | 0.81 |
| Val ACC | 0.89 | 0.91 | 0.88 | 0.91 | 0.88 | 0.92 | 0.90 | 0.87 | 0.81 |
| Test F1 | 0.89 | 0.91 | 0.88 | 0.90 | 0.87 | 0.90 | 0.90 | 0.85 | 0.81 |
| Test ACC | 0.89 | 0.91 | 0.88 | 0.90 | 0.87 | 0.90 | 0.90 | 0.85 | 0.81 |

Table 3: Results of correlation between skin nuance and F1-score and Accuracy

| Dataset | Model | Prior Training | | Posterior Training | | Changes | |
|---|---|---|---|---|---|---|---|
| | | F1-score | Accuracy | F1-score | Accuracy | F1-score | Accuracy |
| UTKFace | EffNet | -0.455 | -0.319 | -0.379 | -0.209 | 0.076 | 0.110 |
| | ResNet | -0.442 | -0.316 | -0.407 | -0.257 | 0.035 | 0.059 |
| | VGG | -0.448 | -0.268 | -0.430 | -0.259 | 0.018 | 0.009 |
| CelebA | EffNet | 0.265 | 0.109 | 0.244 | 0.086 | 0.021 | 0.023 |
| | ResNet | 0.115 | -0.084 | 0.111 | -0.084 | 0.004 | 0.000 |
| | VGG | 0.156 | 0.040 | 0.150 | -0.029 | 0.006 | 0.011 |
| HAM | EffNet | -0.513 | -0.555 | -0.412 | -0.329 | 0.101 | 0.226 |
| | ResNet | -0.629 | -0.424 | -0.533 | -0.355 | 0.096 | 0.069 |
| | VGG | -0.497 | -0.377 | -0.600 | -0.425 | -0.103 | 0.048 |

## 6 DISCUSSION

The nuances of the pigments, which had previously been neglected, were measured by the probability distribution with statistical distance. The results of Bayesian regression exposed the existence of a bias that could not be detected by fairness between groups. It was demonstrated that the correlation between distance and performance was mitigated by the loss function, which re-weighted the difference in skin color as a penalty. The starting epoch to apply the penalty differs depending on the combination of the model and dataset. In this experiment, most combinations succeeded by beginning about 30% of the total training epochs for most combinations. Although Sample 3 in Figure 2 is a monochrome image, it has been annotated by human intuition and classified as 'white'. However, when observing the color alone, it is apparent that it differs from other 'white' skin tones, highlighting the limitations of relying solely on human-assigned labels. This involves consideration beyond mere color perception. Distinctly, our approach focuses exclusively on the skin tone of the image being evaluated, which obviates the need for it to be supplemented by subjective assessments or other extrinsic factors. **This unique perspective has not been explored in prior research, therefore, a direct comparison with existing techniques is not feasible. This underscores the novelty of our method in addressing fairness in image classification by isolating and analyzing the inherent skin tone directly from the image data for the first time.**

### 6.1 FUTURE WORK

There are two possible future tasks for this research. Although this manuscript focused on Wasserstein Distance, it is possible to reduce further performance differences due to individual skin color by investigating various statistical distance methods. The method can also be applicable to image-to-image generation and language-to-image models. The method allows us to evaluate the variation in the skin color range of the generated images.

## 6.2 LIMITATIONS

This proposal requires the identification of skin pixels. The detection of skin pixels relies on existing methods, such as publicly available segment images and landmarks. However, the skin detection mechanism is out of our research scope. It cannot be applied to datasets lacking skin detection methods, such as Fitzpatrick17K Groh et al. (2022; 2021) and Diverse Dermatology Images Daneshjou et al. (2022), in cases where there is no segment data, tiny skin areas, or skin lesions of multiple individuals in a single image.

## 7 CONCLUSION

The performance of models with different skin tones of individuals was assessed by measuring the gradation matrix that skin tones have using statistical distance measures and without categorising skin types. The results demonstrated that biases latent within the same category could be detected. Moreover, by weighting the loss function according to nuanced differences in skin color, the correlation with the target evaluation metric was significantly reduced. In the future, this mechanism could be applied to generative models.

### CODE AVAILABILITY

Made Hidden, as the paper under, is a double-blind review.

### ACKNOWLEDGMENTS

Made Hidden, as the paper under, is a double-blind review.

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

# A APPENDIX

## A.1 PRIOR TRAINING MODEL PERFORMANCCE

This Table 4 provides the performance results of a generic model with a commonly assessed group fairness. In this research, the model was employed for the purpose of Bayesian regression prior distributions.

Table 4: Experiment models and the general performance

| Dataset | UTKFace | | | CelebA | | | HAM | | |
|---|---|---|---|---|---|---|---|---|---|
| Model | EffNet | ResNet | VGG | EffNet | ResNet | VGG | EffNet | ResNet | VGG |
| lr | 1e-5 | 1e-5 | 1e-6 | 1e-6 | 1e-6 | 1e-6 | 1e-5 | 1e-6 | 1e-6 |
| Epochs | 14 | 17 | 24 | 29 | 28 | 24 | 18 | 23 | 33 |
| Val F1 | 0.92 | 0.90 | 0.88 | 0.91 | 0.88 | 0.91 | 0.89 | 0.80 | 0.83 |
| Val ACC | 0.92 | 0.90 | 0.88 | 0.91 | 0.88 | 0.91 | 0.89 | 0.80 | 0.83 |
| Test F1 | 0.91 | 0.90 | 0.88 | 0.91 | 0.87 | 0.90 | 0.88 | 0.78 | 0.82 |
| Test ACC | 0.91 | 0.90 | 0.88 | 0.91 | 0.87 | 0.90 | 0.88 | 0.78 | 0.82 |

# B EQUAL OPPORTUNITY, EQUAL ODDS, DEMOGRAPHIC PARITY FOR CONTINUOUS SENSITIVE ATTRIBUTES USING WASSERSTEIN DISTANCE

In this appendix, we extend the traditional Equal Opportunity fairness constraint to accommodate continuous sensitive attributes by incorporating the Wasserstein Distance (WD). Specifically, we address the challenge of applying fairness metrics to a continuous attribute such as skin tone, where traditional binary or categorical approaches are insufficient.

## B.1 BACKGROUND

The Equal Opportunity criterion Hardt et al. (2016) ensures that the true positive rates are equal across different groups defined by a sensitive attribute $A$. For a binary sensitive attribute, the fairness constraint is expressed as:

$$P\left(\hat{Y} = 1 \mid A = 0, Y = 1\right) = P\left(\hat{Y} = 1 \mid A = 1, Y = 1\right), \tag{6}$$

where $\hat{Y}$ is the predicted label and $Y$ is the true label.

### B.2 EXTENSION TO CONTINUOUS SENSITIVE ATTRIBUTES

When $A$ is continuous (e.g., skin tone measured on a continuous scale), Equation equation 6 is not directly applicable. To address this, we introduce a distance metric that quantifies the difference between different values of $A$ and a reference point $A_0$ (e.g., the lightest skin tone). We use the Wasserstein Distance to measure this difference.

### B.3 WASSERSTEIN DISTANCE WITH DIRECTIONAL SIGNIFICANCE

Let $\mathcal{F}(A)$ denote the cumulative distribution function (CDF) of the sensitive attribute $A$. The Wasserstein Distance between two values $A_0$ and $A_i$ is defined as:

$$\mathcal{D}(A_0, A_i) = \int_{-\infty}^{\infty} |\mathcal{F}(A_0) - \mathcal{F}(A_i)| \, dA \cdot \text{sign}(A_i - A_0), \tag{7}$$

where $\text{sign}(A_i - A_0)$ captures the direction of the difference, indicating whether $A_i$ is greater than or less than $A_0$.

### B.4 REWRITING THE EQUAL OPPORTUNITY CONSTRAINT

We adjust the Equal Opportunity constraint to incorporate the continuous nature of $A$ and the distance metric:

$$\int_{-\infty}^{\infty} \mathcal{D}(A_0, A) \left[ P\left(\hat{Y} = 1 \mid A, Y = 1\right) - P\left(\hat{Y} = 1 \mid A_0, Y = 1\right) \right] dF_{A|Y=1}(A) = 0, \tag{8}$$

where $dF_{A|Y=1}(A)$ is the probability density function of $A$ given $Y = 1$.

### B.5 INTERPRETATION

Equation equation 8 ensures that the weighted difference in true positive rates between any value of $A$ and the reference point $A_0$ integrates to zero over the distribution of $A$ given $Y = 1$. The weighting by $\mathcal{D}(A_0, A)$ accounts for both the magnitude and direction of the difference in the sensitive attribute.

### B.6 DEMOGRAPHIC PARITY

#### B.6.1 BACKGROUND

Demographic Parity (DP) Zafar et al. (2017) is a fairness criterion that requires the predicted outcome $\hat{Y}$ to be independent of the sensitive attribute $A$. For a binary sensitive attribute, DP is defined as:

$$P\left(\hat{Y} = 1 \mid A = 0\right) = P\left(\hat{Y} = 1 \mid A = 1\right). \tag{9}$$

#### B.6.2 EXTENSION TO CONTINUOUS SENSITIVE ATTRIBUTES

When $A$ is continuous, Equation equation 9 is not directly applicable. To extend DP to continuous $A$, we utilize the Wasserstein Distance to measure the difference between different values of $A$ and a reference point $A_0$ (e.g., the lightest skin tone).

#### B.6.3 WASSERSTEIN DISTANCE WITH DIRECTIONAL SIGNIFICANCE

Let $\mathcal{F}(A)$ denote the cumulative distribution function (CDF) of the sensitive attribute $A$. The Wasserstein Distance between two values $A_0$ and $A$ is defined as:

$$\mathcal{D}(A_0, A) = \int_{A_0}^{A} |\mathcal{F}(a) - \mathcal{F}(A_0)| \, da \cdot \text{sign}(A - A_0), \tag{10}$$

where $\text{sign}(A - A_0)$ captures the direction of the difference.

### B.6.4 REWRITING THE DEMOGRAPHIC PARITY CONSTRAINT

We adjust the Demographic Parity constraint to incorporate the continuous nature of $A$ and the distance metric:

$$\int_{-\infty}^{\infty} \mathcal{D}\left(A_0, A\right) \left[P\left(\hat{Y} = 1 \mid A\right) - P\left(\hat{Y} = 1 \mid A_0\right)\right] dF_A(A) = 0, \tag{11}$$

where $dF_A(A)$ is the probability density function of $A$.

### B.6.5 INTERPRETATION

Equation equation 11 ensures that the weighted differences in the probability of a positive prediction between any value of $A$ and the reference point $A_0$ integrate to zero over the distribution of $A$. The weighting by $\mathcal{D}\left(A_0, A\right)$ accounts for both the magnitude and direction of the differences in the sensitive attribute.

## B.7 EQUALIZED ODDS

### B.7.1 BACKGROUND

Equalized Odds (EO) Hardt et al. (2016) requires that both the true positive rates (TPR) and false positive rates (FPR) are equal across groups defined by the sensitive attribute $A$. For a binary-sensitive attribute, EO is expressed as:

$$P\left(\hat{Y} = 1 \mid A = 0, Y = y\right) = P\left(\hat{Y} = 1 \mid A = 1, Y = y\right), \quad \text{for } y \in \{0, 1\}. \tag{12}$$

### B.7.2 EXTENSION TO CONTINUOUS SENSITIVE ATTRIBUTES

To extend EO to a continuous $A$, we again incorporate the Wasserstein Distance to account for differences across the continuous domain.

### B.7.3 REWRITING THE EQUAL ODDS CONSTRAINT

The adjusted EO constraint is given by:

$$\int_{-\infty}^{\infty} \mathcal{D}\left(A_0, A\right) \left[P\left(\hat{Y} = 1 \mid A, Y = y\right) - P\left(\hat{Y} = 1 \mid A_0, Y = y\right)\right] dF_{A|Y=y}(A) = 0, \quad \text{for } y \in \{0, 1\}, \tag{13}$$

where $dF_{A|Y=y}(A)$ is the conditional probability density function of $A$ given $Y = y$.

### B.7.4 INTERPRETATION

Equation equation 13 ensures that the weighted differences in prediction probabilities between any value of $A$ and the reference point $A_0$, conditioned on the true label $Y = y$, integrate to zero over the distribution of $A$ given $Y = y$. This enforces that both TPR and FPR are balanced across the spectrum of the sensitive attribute.

## B.8 IMPLICATIONS

These formulations generalize the Equal Opportunity, Demographic Parity and Equalized Odds criteria to continuous sensitive attributes by:

- Utilizing the Wasserstein Distance to quantify differences across the continuous domain of $A$.
- Incorporating the sign function to maintain the directional significance of these differences.
- Ensuring fairness by balancing the weighted disparities in prediction probabilities across all values of $A$.

