# OpenReview forum: "Evaluating Fairness and Mitigating Bias in Machine Learning: A Novel Technique using Tensor Data and Bayesian Regression"
_ICLR.cc/2025/Conference — ICLR 2025 Conference Withdrawn Submission_

### Official Review · Reviewer_1tG4 · 2024-10-25

**Soundness:** 2
**Presentation:** 2
**Contribution:** 1
**Rating:** 3
**Confidence:** 4

**Summary:**

The paper addresses the conventional approach to skin tone annotation by proposing a novel method that treats skin tone as a continuous variable rather than a categorical classification. The authors leverage the raw values obtained through Individual Typology Angle (ITA) measurements, utilizing these continuous measurements before their traditional conversion into discrete categories. Building upon this continuous representation, they develop a bias mitigation framework that incorporates the distance information derived from these ITA values to create a regularized training loss function. Experiments are conducted on established group fairness benchmark datasets.

**Strengths:**

The approach of treating physical characteristics as continuous variables, rather than discrete categories, is compelling. This applies not only to skin tone, but also to other demographic attributes (eg. age, perceived gender,...) and physical features (eg. hair color, perceived attractiveness, ...). While the idea of adopting continuous representations isn't novel [1,2] and the proposed method applies only for skin tone, the idea of implementing it without requiring annotated data presents an interesting research direction.

[1] Kumar, Neeraj, et al. "Attribute and simile classifiers for face verification." 2009 IEEE 12th international conference on computer vision. IEEE, 2009.

[2] Moeini, Ali, et al. "Regression Facial Attribute Classification via simultaneous dictionary learning." In Pattern Recognition, volume 62, pages 99-113, 2017. DOI: https://doi.org/10.1016/j.patcog.2016.08.031

**Weaknesses:**

1. Although I understand that the scope is to focus on skin tone, the study is a bit limited as the proposed methodology seemingly doesn't transfer to any other attribute of interest (also other attributes may benefit from treating them in a continuous range of values, rather than as categorical variables, eg. "age").
2. The proposed methodology raises several concerns regarding its novelty and effectiveness. The required preprocessing step appears to be a general solution that could be applied to any existing method, rather than a unique contribution. Furthermore, the training process relies heavily on conventional binary classification with regularization, without demonstrating significant innovation. The absence of comparisons with state-of-the-art unfairness mitigation techniques makes it difficult to evaluate the method's relative merits. Most critically, the lack of baseline comparisons leaves readers unable to assess the tangible advantages this approach might offer over existing solutions.
3. The manuscript would benefit from several structural and technical refinements. In terms of organization, the contributions section should be relocated to the end of the introduction. The current list of contributions requires revision: contributions #2 and #3 should be consolidated as they represent a single advancement, while contributions #4 (experimental validation) and #5 (code sharing) should be removed as they represent standard research practices rather than novel contributions. The related works section should conclude with a clear paragraph distinguishing this study from existing literature. Additionally, the paper needs technical cleanup, including addressing various grammatical errors and misspellings, adding a missing reference on line 46, and improving the legibility of Figure 3, which is currently difficult to read.

**Questions:**

See "Weaknesses".

_**Justification of Rating**_

The paper presents one noteworthy concept: the treatment of sensitive attributes as continuous variables rather than discrete categories. This approach is particularly well-suited for skin color, where it can be implemented straightforwardly through pixel value statistics. However, this single contribution, while valuable, is insufficient to warrant acceptance in its current form.
A more comprehensive contribution would develop a framework capable of handling various sensitive attributes as continuous variables (such as age) rather than limiting the scope to skin color alone. The current implementation, while promising in concept, remains too narrow in its application and theoretical development.
Two significant deficiencies further impact the paper's potential acceptance:

1. The absence of comparative analysis against existing methods makes it impossible to evaluate the practical benefits of this approach. Without such benchmarking, the methodology's advantages remain purely theoretical.
2. The paper's organizational structure requires substantial improvement to effectively communicate its contributions and methodology.

These limitations, combined with the narrow scope of the primary contribution, lead me to recommend against acceptance in its current form. However, with expanded scope, rigorous comparative analysis, and improved organization, this work could develop into a significant contribution to the field.

---

### Official Review · Reviewer_yBQL · 2024-10-31

**Soundness:** 2
**Presentation:** 1
**Contribution:** 2
**Rating:** 3
**Confidence:** 4

**Summary:**

This paper introduces a method for measuring fairness and mitigating bias in machine learning models that handle skin color as tensor data rather than traditional categorical labels. The approach leverages probability distributions and Wasserstein Distance, to capture detailed variations in skin tone, allowing for an individualized fairness assessment. The paper proposes a Bayesian regression model that predicts performance outcomes based on these nuanced skin color distributions, rather than on coarse demographic categories. Additionally, the study introduces a training method that mitigates bias through a weighted loss function, penalizing model performance inversely to the predicted fairness distance. This approach aims to reduce latent biases within and across typical group classifications, thus improving fairness in image classification tasks without requiring skin color annotation. The empirical results demonstrate a reduced correlation between skin tone and prediction accuracy.

**Strengths:**

1. Introduces a novel approach to fairness by representing skin color as continuous tensor data, avoiding traditional categorical groupings.
2. Uses Bayesian regression and Wasserstein Distance to capture individual-level fairness without requiring categorical annotations.

**Weaknesses:**

**Insufficient Coverage and Comparison with Related Works:**
The paper does not provide a discussion on dependence-based methods [6-11] or adversarial representation learning approaches [1-5], both of which are established techniques for debiasing machine learning models. While the setting of this study is distinct, the continuous skin tone attribute extracted in the initial phase of this method could also be applied in models handling continuous attributes, aligning with those frameworks.

I have listed some relevant works below that are capable of handling the data type used in your method, providing potential baselines for comparing the proposed approach:


[1] Wang, Tianlu, et al. "Balanced datasets are not enough: Estimating and mitigating gender biases in deep image representations." ICCV, 2019.\
[2] Roy, Proteek Chandan, and Vishnu Naresh Boddeti. "Mitigating information leakage in image representations: A maximum entropy approach." CVPR, 2019.\
[3] Edwards, Harrison, and Amos Storkey. "Censoring representations with an adversary." arXiv, 2015.\
[4] Xie, Qizhe, et al. "Controllable invariance through adversarial feature learning." NeurIPS, 2017.\
[5] Madras, David, et al. "Learning adversarially fair and transferable representations." ICML, 2018.\
[6] Dehdashtian, Sepehr, et al. "Utility-Fairness Trade-Offs and How to Find Them." CVPR, 2024.\
[7] Sadeghi, Bashir, et al. "On characterizing the trade-off in invariant representation learning." TMLR, 2022.\
[8] Sadeghi, Bashir, et al. "Adversarial representation learning with closed-form solvers." ECML-PKDD, 2021.\
[9] Quadrianto, Novi, et al. "Discovering fair representations in the data domain." CVPR, 2019.\
[10] Chzhen, Evgenii, et al. "Fair regression with Wasserstein barycenters." NeurIPS, 2020.\
[11] Jiang, Ray, et al. "Wasserstein fair classification." UAI, 2020.

Without comparison to a baseline from the list above, it may be challenging to accurately assess the performance of the proposed model and validate the paper's claimed contributions.

## Minor Edits
1. There appears to be an unidentified reference in line 46.
2. In line 207, the phrase "Assuming the distributions are IID" may be inaccurate; it seems more likely that "samples are i.i.d." was intended.

**Questions:**

All the questions and suggestions are mentioned in the Weaknesses section.

---

### Official Review · Reviewer_LbvP · 2024-11-12

**Soundness:** 2
**Presentation:** 2
**Contribution:** 1
**Rating:** 3
**Confidence:** 4

**Summary:**

This paper addresses individual fairness when the sensitive attribute is skin color. Most
literature deals with categorical sensitive features, while skin color is a tensor and even
its annotation can be often lacking. The proposed method avoids classifying the color
into categories, and aims to capture fine-grained nuances in fairness. Instead, it
represents it into probability distributions and apply Wasserstein distance, based on
which Bayesian regression with polynomial functions is used to estimate the
performance. Finally, the latent bias is mitigated by reweighting the cross-entropy loss
with the prediction performance (after softmax).

**Strengths:**

1. Extend categorical groups by representing skin color as distributions on which
Wasserstein Distance can be applied. The method is generically applicable to multi-
dimensional and continuous data.

2. A new latent bias mitigation method is proposed for individual fairness that leverages
Bayesian regression estimation of performance.

**Weaknesses:**

1. Although skin color is an important fairness indicator and its continuity fits the
motivation of the paper, it appears a significant limitation to only consider skin
color. There are many other continuous sensitive features, and the paper didn’t
consider in the experiment. Is it because they are too easy and do not unleash
the full power of the method (which can be applied to tensors)? It will be
interesting to see the effectiveness of the proposed method on other continuous
valued attributes.
2. What about using the logit of multi-class or multi-label classification of skin color?
The current color distribution is constructed in an unsupervised fashion. So how
can we guarantee that eventually what is learned/extracted is not targeting other
features of skin, say, coarseness. Although I do agree that color is probably the
most salient feature of skin, does it mean the method has to be hand-tuned for
each domain?
3. The presentation is very unclear in Section 3.2. Is $n$ the batch size that was
set to 1% of the validation dataset? If the distances $d_i$ are all with respect to
$x_0$, then does it mean that the performance of $n$ instances are based on
just one baseline image $x_0$?

**Questions:**

See weaknesses

---

### Note · Authors · 2024-11-26

**Comment:**

First and foremost, we would like to sincerely thank you and the reviewers for the feedback provided on our submission. The suggestions are valuable and have begun to shape the direction of our research.
After careful consideration, we have decided to withdraw our paper from the conference. This decision was not made lightly. Due to the reviewer's request to expand the scope of our experiments to prove the applicability, we realised that it would not be feasible within the discussion period. We intend to address the reviewers' suggestions thoroughly and hope to resubmit an improved version of our research in the future. Thank you very much.

**Withdrawal Confirmation:**

I have read and agree with the venue's withdrawal policy on behalf of myself and my co-authors.